# Phylogenetic Reclassification of *Metarhizium granulomatis* and *Metarhizium viride* Species Complex

**DOI:** 10.3390/pathogens14080745

**Published:** 2025-07-29

**Authors:** Johanna Würf, Volker Schmidt

**Affiliations:** Clinic for Birds and Reptiles, Leipzig University, 04103 Leipzig, Germany; volker.schmidt@vogelklinik.uni-leipzig.de

**Keywords:** clavicipitaceae, RNA polymerase II second largest subunit, translation elongation factor 1 alpha, hyalohyphomycosis, Reptiles

## Abstract

*Metarhizium (M.) granulomatis* and *M. viride* have previously been described as pathogens causing hyalohyphomycosis in various species of captive chameleons and bearded dragons (*Pogona vitticeps*). Previous studies yielded different genotypes of *M. granulomatis* and *M. viride* based on sequencing of the internal transcribed spacer 1-5.8S rDNA (ITS-1-5.8S) and a fragment of the large subunit of the 28S rDNA (LSU). The aim of this study was to clarify the relationships between these genotypes and obtain a more accurate phylogenetic classification by sequencing two different loci of the RNA polymerase II second largest subunit (NRPB2), referred to as RPB1 and RPB2, and the translation elongation factor 1 alpha (EF1α). A total of 23 frozen isolates from 21 lizards, including the first isolates of *M. granulomatis* and *M. viride* from Parson’s chameleons (*Calumma parsonii*), were available for phylogenetic analysis. A total of 13 isolates belonged to the *M. granulomatis* complex and 10 isolates belonged to the *M. viride* complex. Following the amplification and sequencing of the protein-coding genes, the resulting nucleotide sequences were analyzed, trimmed and assembled. These were further analyzed with regard to differences in single-nucleotide polymorphisms (SNPs) and amino acid structure. In consideration of the results of the present analyses, a phylogenetic reclassification is recommended. Three different genotypes of *M. granulomatis* can be distinguished, which can be phylogenetically addressed as subspecies. Six subspecies can be distinguished regarding *M. viride.*

## 1. Introduction

The importance of fungi within the family Clavicipitaceae, specifically the genus *Metarhizium* (*M.*), is constantly increasing. In regard to reptile medicine, diseases caused by *Metarhizium* have so far been described in several species of lizards, turtles, and tortoises, as well as in crocodiles [1,2,3]. A pivotal element in the pathogenesis of these fungal species is the significant reliance of the immune system in poikilothermic animals on the prevailing husbandry conditions, including temperature, within their environment [4]. Humidity, diet, and UV light conditions have a constant impact on reptilian health [5].

*Metarhizium granulomatis* and *M. viride* are known to cause hyalohyphomycosis in captive lizards [1,2]. In particular, *M. granulomatis* and *M. viride* have been described as primary pathogens, affecting chameleons and central bearded dragons and causing fungal dermatitis, granulomatous glossitis, pharyngitis, and disseminated visceral mycosis [6,7]. However, concurrent colonization and/or infection with both species in lizards and chameleons has also been described, emphasizing the importance of specifically identifying these fungal pathogens [8]. The most recent research results indicate that these keratinophilic fungi may constitute species complexes consisting of five different genotypes of *M. granulomatis* [6] and four different genotypes for *M. viride* [7]. There appears to be a strong correlation between *M. granulomatis* genotype A and the occurrence of dermatitis [6]. Conversely, findings suggest that *M. granulomatis* genotypes B–E are linked to the development of pharyngitis and glossitis [6]. However, ribosomal genes of the internal transcribed spacer1-5,8s (ITS1-5,8S) and a fragment of the large subunit of the 28S rDNA (LSU) have been sequenced [6,7], constituting data that does not appear to be sufficient for phylogenetic studies on the family Clavicipitaceae [9,10]. Although ITS and LSU are extensively utilized for fungal barcoding, they frequently prove ineffective in resolving closely related taxa. In contrast, the utilization of single-copy, protein-coding genes such as NRPB2 and EF1α has been demonstrated to facilitate enhanced phylogenetic resolution and enable more precise discrimination of closely related genotypes [9,10].

The objective of this study was to determine whether the previously described genotypes of *M. granulomatis* and *M. viride* [6,7] can be underlined and differentiated more comprehensively through the integration of NRPB2 (RPB1, RPB2) and EF1α. A comprehensive evaluation of single-nucleotide polymorphisms (SNPs) was conducted, followed by a translation of nucleotides into amino acid patterns. This approach enabled the formulation of conclusions that contribute to a more profound comprehension of the phylogenesis inside the *M. granulomatis* and *M. viride* complexes. The significance of accurate phylogenetic classification, therefore, lies in its potential to reveal differences in host specificity, virulence, and pathogenesis, thereby potentially preventing infections.

## 2. Materials and Methods

### 2.1. Fungal Isolates

Fungal isolates were collected between 2010 and 2023 from lizards (*n* = 21) presented to the Clinic for Birds and Reptiles, University Leipzig, Germany. The isolates were derived from the following species: veiled chameleons (*Chameleo calyptratus*) (*n* = 11), panther chameleons (*Furcifer pardalis*) (*n* = 4), Parson’s chameleons (*Calumma parsonii*) (*n* = 3), central bearded dragons (*Pogona vitticeps*) (*n* = 2), and a carpet chameleon (*Furcifer lateralis*) (*n* = 1). Specifically, the isolates were obtained from the throat (*n* = 9), the cloaca (*n* = 7), the tongue (*n* = 4), liver (*n* = 2), fecal samples (*n* = 2), or the rectum (*n* = 1). The isolates were stored in ROTI^®^Store yeast cryotubes (Carl Roth GmbH & Co.KG, Karlsruhe, Germany) at a temperature of −32 °C until utilization.

Genomic DNA was purified using a DNeasy blood-and-tissue kit (Qiagen, Hilden, Germany), following the protocols given by the manufacturer. In the following steps, species affiliation was proven via ribosomal DNA sequencing [2,6,7]. Regarding *M. granulomatis*, genotype A was obtained from Parson’s chameleons (*n* = 2) and a panther chameleon, while genotype B isolates were obtained from four chameleon species, including one Parson’s chameleon with co-isolation of genotype A. Genotypes C and D were only obtained from veiled chameleons (*n* = 2, each). One of these veiled chameleons exhibited genotypes B and C.

Regarding *M. viride*, genotype A was derived from veiled chameleons (*n* = 2); genotype B was obtained from a central bearded dragon and a panther chameleon (one each); and genotype C was obtained from one veiled chameleon, one panther chameleon, and one Parson’s chameleon. Genotype D (*n* = 1) was obtained from a central bearded dragon. In addition, two isolates with previously unpublished genotyping were obtained from veiled chameleons (*n* = 2). The purified genomic DNA of these isolates was frozen at −32 °C in Eppendorf tubes until further examination.

### 2.2. PCR and DNA Sequencing

The forward and reverse primers employed in this study were selected based on previous phylogenetic attempts to reclassify *Metarhizium* spp. and are listed in Table 1 [11,12,13,14].

The DNA quality of the frozen samples was determined via nanodrop spectrophotometry using a NanoPhotometer^®^ NP80 (Implen GmbH, Munich, Germany) (Appendix A). DNA-free controls were used as negative samples. In addition, two to three PCR replicates were employed for each sample. In cases of co-occurring genotypes, separation was carried out during the cultivation process. Afterwards, the morphologically different colonies were frozen and labeled as required [8]. All of the samples were analyzed according to described protocols [6], focusing on partial sequences of three gene loci: RPB1, RPB2, and EF1α [11,12,13,14]. There was a repetition in the PCR of RPB1 due to a change in the annealing temperature from 58 °C to 60 °C. After the samples were amplified, detection was carried out using gel electrophoresis. The PCR products were further processed via polyethylene glycol precipitation [15]. Sequencing was conducted via bidirectional Sanger sequencing, with subsequent sequence quality verification performed using the Phred base-calling program for DNA sequence traces, provided by a commercial DNA-sequencing service (Faculty of Medicine, Central Functional Area: DNA Technologies/Sequencing, Leipzig, Germany). A volume of 10 μL of the amplified sample, in conjunction with the primers, was employed in accordance with the previously delineated protocol.

### 2.3. Analyzing Data According to Phylogeny

The alignment lengths for each gene locus were 819 bp for RPB1 and 802 bp for RPB2. With respect to EF1α, these lengths were 729 bp and 723 bp for *M. granulomatis* and *M. viride*, respectively. As a next step, all sequences were edited into a multilocus dataset, combining EF1α (non-coding and CDS), RPB1, and RPB2 resulting in a total length of 2350 bp using MEGA12 (Molecular Evolutionary Genetics Analysis Version 12 for adaptive and green computing [16]). Comparison with already available data in the GenBank database was carried out according to the BLAST program (http://www.ncbi.nlm.nih.gov/BLAST/, accessed on 5 March 2025) [17].

Furthermore, the isolates were evaluated in the four following datasets: RPB1, RPB2, EF1α, and the multilocus dataset. First, all the isolates were analyzed regarding single-nucleotide polymorphisms (SNPs) on a nucleotide base. The EF1α as well as the multilocus dataset also contained coding and non-coding regions of EF1α, as so including these can lead to a better analysis at the molecular genetic level. The phylogeny was inferred using the maximum likelihood method and the Jukes–Cantor model [18] of nucleotide substitutions and the tree with the highest log likelihood (−3675.87) is shown. The percentage of replicate trees in which the associated taxa clustered together (1000 replicates) is shown next to the branches [19]. The analytical procedure encompassed 28 coding nucleotide sequences using 1st positions with 843 positions in the final dataset.

The isolates then were translated into amino acid patterns to detect any differences in branching. This led to a trimming of EF1α for *M. granulomatis* and *M. viride* in order to focus on CDS regions. The resulting length for EF1α was 257 bp, and the length of the multilocus dataset was 1878 bp. Evolutionary history and potential similarity were derived using the maximum likelihood method based on the Jones–Taylor–Thornton model [20]. The percentage of replicate trees in which the associated taxa clustered together (1000 replicates) is shown next to the branches [19]. The evolutionary-rate differences among sites were modeled using a discrete Gamma distribution across 5 categories (+G, parameter = 2.5938). The analytical procedure encompassed 28 coding nucleotide sequences using 1st positions, with 843 positions in the final dataset.

To create additional outgroup taxa, isolates of the *Metarhizium anisopliae* group (informally designated as the “PARB” clade) [21], such as *M. anisopliae*, *M. brunneum,* and *M. robertsii,* were added. A member of the family Clavicipitaceae that was used was *M. acridum*. Also, *Beauveria bassiana* was added to act as a root. To create comparative isolates for the multilocus analysis, sequences of RPB1, RPB2, and EF1α from the outgroup taxa were taken from the NCBI GeneBank database (http://www.ncbi.nlm.nih.gov). They were then assembled as described earlier. *M. anisopliae* consisted of OK336701.1 [Du, 2021, unpublished [22]] for RPB1 and RPB2 and of DQ463996.2 [14] for EF1α. *M. brunneum* consisted of EU248854.1 [23] (RPB1, EF1α) and XM_014688283.1 [24] (RPB2). *M. robertsii* consisted of DQ468368.1 [23] (RPB1), XM_007819877.1 [25] (RPB2), and DQ463994.2 [14] (EF1α). *M. acridum* consisted of KM527862.1 [Zhang, 2014, Unpublished [26]] (RPB1), XM_066120576.1 [27] (RPB2), and MK391183.1 [28] (EF1α). *Beauveria bassiana* was assembled from MN401612.1 [29] (RPB1), LC812020.1 [Yang et al., 2024, unpublished [30]] (RPB2), and MN026878.1 [Dalla Nora, 2019, Unpublished [31]] (EF1α).

To underline the results obtained using the maximum likelihood method, pairwise distance was computed via the MEGA program (v12) using the Maximum Composite Likelihood (MCL) approach.

## 3. Results

### 3.1. RNA Polymerase II Second Largest Subunit (NRPB2) of M. granulomatis

The sequencing of the first locus (RPB1) of NRPB2 revealed no differences between *M. granulomatis* isolate types A, C, and D, which exhibited 100% identity with respect to *M. granulomatis* [NCBI Acc.-Nr. KJ398688.1 [10]]. The sole discernible distinction between *M. granulomatis* type B and other types was a single-nucleotide polymorphism. However, translation into an amino acid sequence showed 100% identity of all the *M. granulomatis* isolates as well as to the additional outgroup taxa of *M. anisopliae* [NCBI Acc.-Nr. OK336701.1 [Du, 2021, unpublished [22]]], *M. brunneum* [NCBI Acc.-Nr. EU248934.1 [23]], *M. robertsii* [NCBI Acc-Nr. DQ468368.1 [23]], and *M. acridum* [NCBI Acc.-Nr. KM527862.1 [Zhang, 2014, Unpublished [26]]] (Appendix A).

Analyzing the nucleotide structure of the second locus (RPB2) of the NRPB2 of *M. granulomatis* isolates allowed us to observe 11 variable sites. In total, there were six SNPs regarding genotype B; four regarding genotypes C and D, which were 100% identical; and only one SNP regarding genotype A. Genotype A showed 99.12–99.38% of identity with respect to other isolates; isolates of genotype B showed an identity of 98.74–99.12% with respect to others. Isolates of genotype C and D showed an identity of 98.74–99.38%. None of these differences had an impact on amino acid sequences, so all the isolates of *M. granulomatis* had an amino acid identity of 100%. They also clustered with the additional outgroup taxa of *M. anisopliae* [NCBI Acc.-Nr. OK336701.1 [Du, 2021, unpublished [22]]], *M. brunneum* [NCBI Acc.-Nr. XM_014688283.1 [24]], *M. robertsii* [NCBI Acc.-Nr. XM_007819877.1 [25]], and *M. acridum* [NCBI Acc.-Nr. XM_066120576.1 [27]], also with an amino acid identity of 100% (Appendix A).

### 3.2. RNA Polymerase II Second Largest Subunit (NRPB2) of M. viride

*Metarhizium viride* was found to contain 11 variable sites in RPB1. The isolates of genotypes A and D and the two isolates of previously unpublished genotype differed by one single nucleotide each. Translation into amino acids, however, showed that they were 100% identical, with an identity of 100% with respect to *M. viride* [NCBI Acc.-Nr. KJ398717 [10]]. Genotype C could be differentiated from the other isolates via three SNPs. This resulted in a nucleotide identity of 99.71% to genotype A isolates and the ones of unpublished genotype, as well as 99.64% to the genotype D isolate. Furthermore, two isolates, which were labeled as *M. viride* genotype B, could be distinguished from each other based on four single-nucleotide polymorphisms, resulting in an identity of 99.71%. These differences were followed by one change in amino acid structure. One of these isolates [NCBI Acc.-Nr. PV231550] showed an identity of 98.18–99.41%, while the other isolate [NCBI Acc.-Nr. PV231551] had an identity of 98.34–99.49% compared to the remaining *M. viride* isolates. *M. granulomatis* [NCBI Acc.-Nr. KJ398688.1 [10]] exhibited an identity of 98.88% with respect to *M. viride* [NCBI Acc.-Nr. KJ398717 [10]]. Between the different genotypes of *M. viride* and *M. granulomatis* considered in this study, an identity of between 98.51 and 98.98% was observed.

There were a total of nine variable sides in the RPB2, resulting in an identity of 99.25–99.50% among all the isolates. Furthermore, no amino acid sequence differences were found, so there was an identity of 100% between all genotypes. Between the *M. granulomatis* and *M. viride* isolates, 97.15–97.71% identity was observed, while the amino acid sequences were 100% identical.

### 3.3. Translation Elongation Factor 1 alpha (EF1α)

The EF1α nucleotides of *M. granulomatis* revealed the presence of 26 variable sites, 22 of which were situated within non-coding regions. These SNPs resulted in a diverse presentation of identity regarding the *M. granulomatis* genotype B isolates, spanning from 98.75% [NCBI Acc.-Nr. PV231586] to over 99.17% [NCBI Acc.-Nr. PV231585, PV231588] and 100% [NCBI Acc.-Nr. PV231584, PV231587]. The *M. granulomatis* genotype A isolates showed an identity of 100%. Furthermore, as no SNPs were detected between the isolates of *M. granulomatis* genotype C and D, they were determined to have an identity of 100%.

Only four of the variable sites were located on the CDS regions. Out of these SNPs, one was observed in isolates that belonged to *M. granulomatis* genotype A. This also had an impact on the amino acid sequence, resulting in the *M. granulomatis* genotype A isolates clustering separately from others with an identity of 98.80%, and an identity of 100% with respect to *M. granulomatis* [NCBI Acc.-Nr. MH619511.1 [2]]. Furthermore, there were two SNPs in the CDS nucleotide sequence of the *M. granulomatis* genotype B isolates, two regarding VS14512 [NCBI Acc.-Nr. PV231585] and VS18409 [NCBI Acc.-Nr. PV231588] and one regarding VS15807 [NCBI Acc.-Nr. PV231586]. These SNPs did not translate into further changes, so all of the other *M. granulomatis* isolates clustered together with an identity of 100% (Appendix A).

The EF1α of *M. viride* isolates revealed a total of 17 variable sites. This resulted in an identity of 98.29–100% of all the isolates with respect to *M. viride* [NCBI Acc.-Nr. MH619515.1 [2]]. Two *M. viride* genotype A isolates presented an identity of 98.73% to the *M. viride* genotype D isolate [NCBI Acc.-Nr. PV231592]. The two isolates designated as *M. viride* genotype B exhibited an identity of 99.16% to each other.

Only one of the variable sites mentioned was located in the CDS region of the EF1α. This SNP belonged to the sequence of the genotype A isolates. No amino acid sequence impact was observed, so all isolates clustered together. Maximum likelihood analyses confirmed a 100% identity with respect to *M. viride* [NCBI Acc.-Nr. MH619515.1 [2]].

A total of 85 variable sites were identified between the EF1α of *M. granulomatis* and *M. viride*, resulting in an identity of 94.24–96.08% regarding the different species. Eight of the single-nucleotide polymorphisms were located on the coding regions. By focusing on them, all the isolates apart from *M. granulomatis* genotype A, had an identity of 100%. Isolates of *M. brunneum* [NCBI Acc.-Nr. EU248854.1 [23]], *M. robertsii* [NCBI Acc.-Nr. DQ463994.2 [14]], *M. anisopliae* [NCBI Acc.-Nr. DQ463996.2 [14]], and *M. acridum* [NCBI Acc.-Nr. MK391183.1 [28]] clustered phylogenetically apart as additional outgroup taxa, resulting in an identity of 96.36–97.58%.

### 3.4. Multilocus Analysis

A total of 38 variable sites were identified through an evaluation of the single-nucleotide polymorphisms present within the multilocus dataset for *M. granulomatis*.

The isolates belonging to *M. granulomatis* genotype A were identical, as there were no differences in nucleotides among them. In addition, they revealed an identity of 99.25–99.46% with respect to the other isolates. One variable site in the isolates of *M. granulomatis* genotypes C and D resulted in an identity of 99.98%. As this SNP was located in the non-coding region of EF1α, no impact on amino acid sequence was observed. Furthermore, the isolates belonging to *M. granulomatis* genotype B showed eight variable sites, which were also located on EF1α. These variable sites facilitated differentiation within this genotype. VS15807 exhibited five SNPs; VS18415, VS17813 and VS17810 exhibited three SNPs; and VS14512 and VS18409 exhibited only one SNP. These results were confirmed by creating a phylogenetic tree and observing the result of branching. The identity among the isolates of genotype B was 99.83–100%.

However, only two of the variable sites were located in the CDS regions. These did not have any impact on the amino acid sequence, as isolates belonging to *M. granulomatis* genotype B had an amino acid identity of 99.90%. Concerning the amino acids, *M. granulomatis* genotype A showed an identity of 98.69–99.02% with respect to the other isolates, while *M. granulomatis* genotype C showed an identity of 98.35–99.02% (Figure 1).

Single-nucleotide polymorphisms of *M. viride* resulted in a total of 41 variable sites. Consequently, six different branches were built by constructing maximum likelihood trees. *M. viride* genotype A isolates VS17602 and VS10221, which were 100% identical, differed from VS2302, formerly named *M. viride* genotype D. This conclusion was based on 11 changes in nucleotide structure and therefore resulted in an identity of 99.52%. As four of these SNPs were located in the coding regions, VS2302 therefore contained three different amino acids. Furthermore, this specific *M. viride* genotype showed an identity of 98.18–98.85% with respect to the remaining isolates and clustered separately. The two isolates formerly named *M. viride* genotype B (VS8419, VS16620) revealed a total of 12 variable sites, resulting in an identity of 99.49%. As four of these SNPs were located in the coding regions, the isolates could be distinguished from each other based on three amino acids. This was also proven by the clustering in the phylogenetic tree.

The isolates belonging to *M. viride* genotype C revealed an identity of 98.97–99.45% among each other. The ones with previously unpublished genotypes, which were 100% identical to each other, showed an identity of 99.10–99.49% with respect to the other isolates. Regarding amino acid sequences, differentiation within six different branches was once again possible. This was a result of the 24 variable sites, which were located in the coding regions and could be translated into 20 differences in amino acid structure among the isolates. (Figure 2).

All the *M. granulomatis* isolates could be distinguished from the *M. viride* isolates with a nucleotide identity of 96.50–96.93%. Once more, including *M. granulomatis* or *M. viride* isolates from the BLAST dataset for comparison was rendered unfeasible, by the paucity of extant published data. A maximum likelihood analysis further confirmed that the earlier described PARB clade is distinct from the *M. granulomatis* and *M. viride* isolates, with the *M. granulomatis* isolates having an identity of 87.38–88.08%. The *M. viride* isolates showed an identity of 86.94–87.79% with respect to the outgroup taxa.

### 3.5. Isolates and Associated Findings

Further evaluation concerning the source of isolation revealed that the isolates belonging to *M. granulomatis* genotypes A and B could all be obtained from cloacal swabs (*n* = 7). In addition, two isolates of *M. granulomatis* genotype B were found in the throat of the lizards sampled. When examining the isolates of *M. granulomatis* genotype C and *M. viride*, the origin of isolation was found to exhibit more diversity. An overview of isolates, host species, origin of isolation and associated pathological findings is shown in Table 2.

A pathological examination was performed on 7 of the 23 lizards (30.4%) sampled. One isolate of *M. granulomatis* genotype A and one of *M. granulomatis* genotype B were isolated from a Parson’s chameleon with diphtheroid colorectitis. However, due to severe bacterial overgrowth, it remains unclear whether the fungi were pathogens or accidental findings. Nevertheless, granulomatous colorectitis caused by *M. granulomatis* genotype C was diagnosed in a veiled chameleon (Appendix A). Three isolates of *M. granulomatis* genotype B were the cause of petechial hemorrhages on the tongue of three different chameleon species. Furthermore, *M. granulomatis* genotype C led to the same finding in veiled chameleons (*n* = 2). *M. viride* genotype A and one isolate of *M. viride* genotype C were found to be the causes of granulomatous glossitis in two veiled chameleons. Furthermore, one isolate belonging to *M. viride* genotype D and one belonging to *M. viride* genotype B were the cause of granulomatous mycotic pharyngitis in central bearded dragons (*n* = 2). Associated pathological findings were lacking for five of the animals presented (21.7%) (Table 2).

## 4. Discussion

In the present study, the evolutionary relationships between fungal isolates of *M. granulomatis* and *M. viride* were investigated. These isolates were obtained from five different lizard species and six different sampling sites. This study presents the first isolates of *M. granulomatis* and *M. viride* obtained from Parson’s chameleons. In addition, the presence of petechiae on the tongue, or necrosis of the caudal tip, was observed in panther chameleons and a carpet chameleon infected with *M. granulomatis*. These findings suggest that *M. granulomatis* may have a broader host range and pathogenicity, as previously described [6].

### 4.1. Phylogenetic Analysis of M. granulomatis Using a Multilocus Approach

By sequencing of the SSU, ITS1-5.8S, and LSU, Schmidt et al. [6] yielded five different genotypes of *M. granulomatis*. In contrast, the multilocus dataset presented here yielded six different genotypes of *M. granulomatis*, which resulted in three different amino acid sequences. The amino acid sequences of these isolates exhibited a similarity of less than 98.7%. A threshold for species separation for bacterial species of ≥98.7–99% was proposed by Stackebrandt and Ebers [32]. No such threshold value has been defined for fungi. Nevertheless, the available data can be used as a guide. Therefore, it may be appropriate to categorize *M. granulomatis* A-C as three distinct subspecies of *M. granulomatis*. Even though the nucleotide sequences of the *M. granulomatis* B isolates showed differences, especially in the non-coding regions, their identity was >98.7%. As a result of the inability to distinguish between *M. granulomatis* subspecies C and D described herein, reclassifying these isolates as a single subspecies is recommended.

### 4.2. Phylogenetic Analysis of M. viride Using a Multilocus Approach

By sequencing of the SSU, ITS1-5.8S and LSU, Schmidt et al. [7] yielded four different genotypes of *M. viride*. In contrast, the multilocus dataset presented here yielded six genotypes of *M. viride*. The isolate formerly labeled as *M. viride* genotype D (VS2302) is genetically very similar to the isolates of genotype A. At this point, one should speak of genetic divergence owing the results of the Maximum Composite Likelihood (MCL) approach. However, since the percentage of similarity to the other isolates of *M. viride* was below 98.7%, it is suggested to reclassify the isolate VS2302. Potentially being renamed as *M. viride* genotype A1, it would appear to be a genetic strain within genotype A. The same applies to the previously mentioned isolates of *M. viride* genotype B (VS8419, VS16620). The results of the multilocus approach also underscore that they are genetically distinct and therefore support a subdivision within *M. viride* genotype B. Furthermore, the isolates belonging to *M. viride* genotype C clustered together in phylogenetic trees built on nucleotides as well as amino acids. Regarding amino acids, their similarity to the other isolates was below the threshold value used for comparison. In conclusion, they comprised one group of subspecies. The isolates of the previously unpublished genotype (VS17601, VS17615) also clustered separately from the others regarding nucleotide and amino acid sequences. The proposal is to refer to them as *M. viride* genotype E.

### 4.3. Evaluation of the Phylogenetic Analysis Using NRPB2 (RPB1)

The first locus (RPB1) of NRPB2 was not target-orientated to yield different genotypes of *M. granulomatis*. No major differences in nucleotide sequences were detected. Only the *M. granulomatis* genotype B isolates clustered separately. Regarding amino acid sequences, all the *M. granulomatis* genotypes as well as the additional outgroup taxa clustered together.

Four genotypes of *M. viride* could be distinguished regarding RPB1. The Maximum Composite Likelihood (MCL) approach showed that isolates of *M. viride* genotypes A, D and the two isolates with previously unpublished genotype could be distinguished using this method. One isolate previously described as *M. viride* genotype D [NCBI Acc.-Nr. PV231549] clustered separately from the other isolates of *M. viride* genotype A regarding nucleotide sequences. Nevertheless, differentiation based on amino acid sequence was not possible. The *M. viride* genotype C isolates showed differences in nucleotide and amino acid sequences. Therefore, it clustered separately from the other isolates. Furthermore, there were significant differences in nucleotides that also translated into the amino acid sequences of the two *M. viride* genotype B isolates. This information could lead to a potential renaming of the isolates to *M. viride* genotype B1 [NCBI Acc.-Nr. PV231550] and *M. viride* genotype B2 [NCBI Acc.-Nr. PV231551], with *M. viride* genotype B1 having the lowest percentage of similarity with respect to the other isolates.

Notwithstanding, a clear distinction between *M. viride* and *M. granulomatis* genotypes could be made using this method.

### 4.4. Evaluation of the Phylogenetic Analysis Using NRPB2 (RPB2)

Regarding RPB2 nucleotides, the sequence similarity among the isolates of *M. granulomatis* and *M. viride* was >98.7%. *M. granulomatis* could be distinguished from *M. viride* with an identity of <98.7%. Regarding amino acids, *M. granulomatis*, *M. viride,* and the PARB-clade clustered together, as there were no significant differences in the CDS. In conclusion, sequencing of the RPB2 alone is not useful for analyzing the phylogenetic diversity of *M. granulomatis* and *M. viride* isolates.

### 4.5. Evaluation of the Phylogenetic Analysis Using EF1α

Similar results were obtained when analyzing the EF1α dataset. Even though the nucleotide sequences, including coding and non-coding regions, exhibited diversity among all the isolates, clear phylogenetic differentiation was not possible. The only isolates showing differences in amino acid sequences and therefore clustering apart from the others were the ones belonging to *M. granulomatis* genotype A. However, the amino acid sequence of EF1α showed more considerable variation among the PARB-clade isolates in comparison to *M. granulomatis* and *M. viride*.

## 5. Conclusions

Finally, after having sequenced and analyzed the data collected in this study, it can be postulated that an individual analysis of RPB1, RPB2, and EF1α is not sufficiently accurate for the specification of *M. granulomatis* and *M. viride*. However, a multilocus analysis of NRPB2 and EF1α represents a comprehensive and meaningful approach to differentiate between species complexes within *M. granulomatis* and *M. viride* as well as defining subspecies within these complexes. By using this method, the genotypes *M. granulomatis* A-D, previously postulated by Schmidt et al. [6], could be reduced to *M. granulomatis* A-C. Furthermore, the genotypes *M. viride* A-D, previously postulated by Schmidt et al. [7], could be subdivided into *M. viride* A, B1, B2, and C. In view of the genetic similarity exhibited by *M. viride* A and *M. viride* D, it is recommended that the designation *M. viride* genotype A1 be employed. Two new previously unclassified isolates could be referred to as *M. viride* genotype E based on the results of a phylogenetic analysis. The present findings serve to corroborate the hypothesis that an analysis of ITS and LSU can be usefully complemented by a multilocus analysis of NRPB2 and EF1a, providing a more profound insight into the phylogenetic divergence of *Metarhizium* spp.

## Figures and Tables

**Figure 1 pathogens-14-00745-f001:**
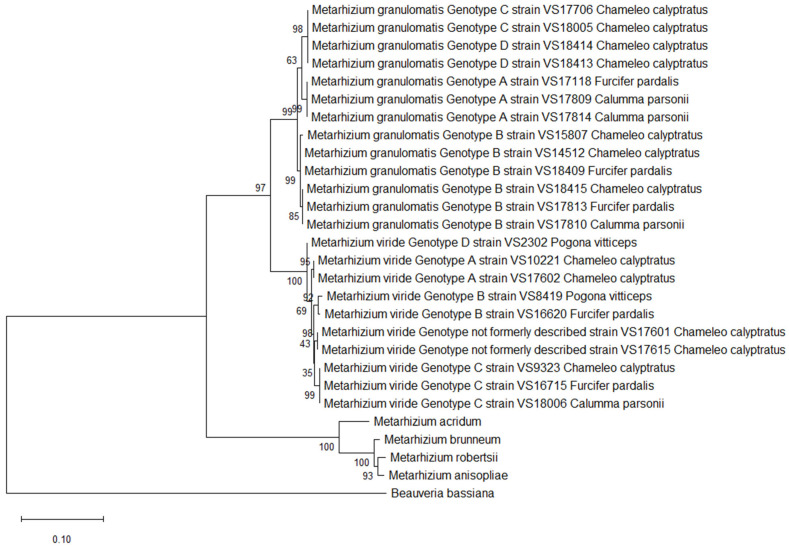
Phylogeny inferred from the analysis of two loci of NRPB2 and coding, as well as non-coding regions of EF1α from 13 isolates of the *Metarhizium* (*M.*) *granulomatis* complex and 10 isolates of the *M. viride* complex, assembled into a multilocus dataset. Analysis was conducted based on nucleotide structures. Other isolates of Claviciptitaceae (Sordariomycetes: Hypocreales) and *Beauveria bassiana* were added as additional outgroup taxa. For this purpose, reference sequences of the two loci of NRPB2 and EF1α along with accession numbers were taken and assembled from the NCBI GeneBank database (http://www.ncbi.nlm.nih.gov).

**Figure 2 pathogens-14-00745-f002:**
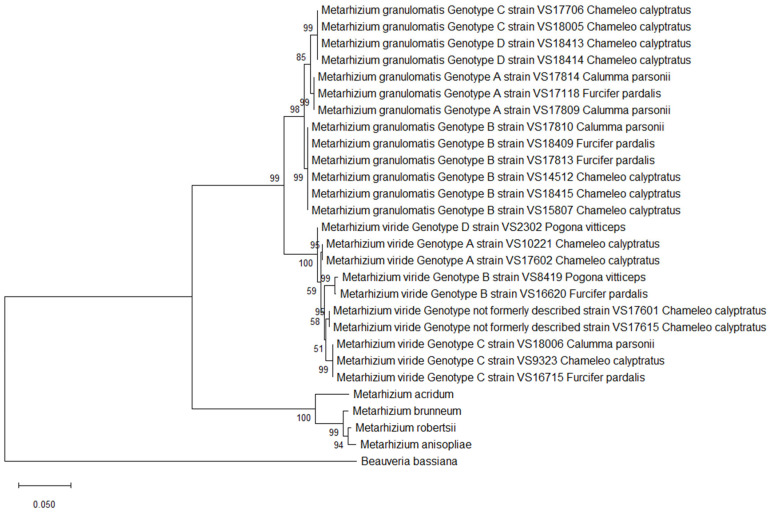
Phylogeny inferred from the analysis of two loci of the NRPB2 and CDS regions of EF1α from 13 isolates of *Metarhizium* (*M.*) *granulomatis*–complex and 10 isolates of *M. viride*–complex, assembled into a multilocus dataset. Analysis was conducted based on amino acid sequences. Other isolates of Claviciptitaceae (Sordariomycetes: Hypocreales) and *Beauveria bassiana* were added as additional outgroup taxa. For this purpose, reference sequences of the two loci of NRPB2 and EF1α along with accession numbers were taken and assembled from the NCBI GeneBank database (http://www.ncbi.nlm.nih.gov).

**Table 1 pathogens-14-00745-t001:** Primers used for the amplification and sequencing of two different loci of NRPB2 (RPB1, RPB2) and the translation elongation factor 1 alpha (EF1α) of *Metarhizium* (*M*.) *granulomatis* and *M*. *viride*.

Target [References]	Forward/Reverse Primers	Annealing Temperature [°C]	Amplicon Size [bp]
RPB1 [11,12]	frPB2-5f2 (5′ GGG GWG AYC AGA AGA AGG C 3′)/ frPB2-7cR (5′ CCC ATR GCT TGY TTR CCC AT 3′)	60	1000
RPB2 [11,13]	fRPB2-7cF (5′ ATG GGY AAR CAA GCY ATG GG 3′)/ fRPB2-11aR (5′ GCR TGG ATC TTR TCR TCS ACC 3′)	58	700
EF1α [14]	5TEF_EF1T (5′ ATG GGT AAG GAR GAC AAG AC 3′)/ 5TEF_EF2T (5′ GGA AGT ACC AGT GAT CAT GT 3′)	58	800

**Table 2 pathogens-14-00745-t002:** Summary of *Metarhizium* (*M.*) *granulomatis* and *M. viride* genotypes (GTs) and isolates identified via sequence analysis of ribosomal DNA, host species, origins of isolates, associated pathological findings, and/or additional disease.

GT	Isolates (NCBI Acc.-Nr.)	Host Species	Origin of Isolation	Associated Pathological Findings and/or Additional Disease
G A	VS17118 *^a^* (PV231534, PV231557, PV231580)	Panther chameleon (*Furcifer pardalis*)	throat and cloaca	Tail tip necrosis
	VS17809 * (PV231535, PV231558, PV231581) VS17814 *^a^* (PV231536, PV231559, PV231582)	Parson’s chameleon (*Calumma parsonii*) (*n* = 2)	cloaca (*n* = 2)	Obstipation, diphtheroid colorectitis; NAF

G B	VS17813 *^a^* (PV231537, PV231560, PV231583)	Carpet chameleon (*Furcifer lateralis*)	cloaca	Petechiae on tongue
	VS18415 *^,#*,a*^ (PV231538, PV231561, PV231584), VS14512 *^a^* (PV231539, PV231562, PV231585), VS15807 (PV231540, PV231563, PV231586)	Veiled chameleon (*Chameleo calyptratus*) (*n* = 3)	throat and cloaca (*n* = 2), throat (*n* = 1)	Petechiae on tongue (*n* = 2), hematochezia, tongue prolapse; granulomatous mycotic pharyngitis; mycobacteriosis
	VS17810 * (PV231541, PV231564, PV231587)	Parson’s chameleon	cloaca	Obstipation, diphtheroid colorectitis
	VS18409 (PV231542, PV231565, PV231588)	Panther chameleon	cloaca	Petechiae on tongue

G C	VS17706 (PV231545, PV231568, PV231591), VS18005 *^a^* (PV231546, PV231569, PV231592)	Veiled chameleon (*n* = 2)	rectum (*n* = 1), throat (*n* = 1)	Hematochezia, petechiae on tongue; egg yolk serositis, granulomatous mycotic colorectitis; squamous cell carcinoma

G D	VS18413 *^a^* (PV231543, PV231566, PV231589), VS18414 *^,#,*a*^ (PV231544, PV231567, PV231590)	Veiled chameleon (*n* = 2)	tongue (*n* = 1), throat and cloaca (*n* = 1)	Hematochezia, petechiae on tongue, coccidiosis; tongue prolapse

V A	VS17602 *^a^* (PV231547, PV231570, PV231593), VS10221 (PV231548, PV231571, PV231594)	Veiled chameleon (*n* = 2)	tongue (*n* = 1), liver (*n* = 1)	Granulomatous mycotic glossitis; visceral mycosis

V B	VS8419 (PV231550, PV231573, PV231596) VS16620 *^a^* (PV231551, PV231574, PV231597)	Central bearded dragon, Panther chameleon	throat and liver, fecal sample	Granulomatous mycotic pharyngitis, visceral mycosis; NAF

V C	VS9323 *^a^* (PV231552, PV231575, PV231598)	Veiled chameleon	tongue	Granulomatous mycotic glossitis, tongue paralysis
	VS16715 *^a^* (PV231553, PV231576, PV231599)	Panther chameleon	throat	NAF
	VS18006 (PV231554, PV231577, PV231600)	Parson’s chameleon	tongue	NAF

V D	VS2302 *^a^* (PV231549, PV231572, PV231595)	Central bearded dragon (*Pogona vitticeps*)	throat	Granulomatous mycotic pharyngitis

V not formerly described	VS17601 *^a^* (PV231555, PV231578, PV231601), VS17615 *^a^* (PV231556, PV231579, PV231602)	Veiled chameleon (*n* = 2)	throat (*n* = 1); fecal sample (*n* = 1)	Petechiae on tongue, conjunctivitis; NAF

*^a^* No pathological exam was performed. *^,#^ isolated simultaneously from one animal and identical location. NAF, no associated pathological finding. “G” = Metarhizium granulomatis, “V” = Metarhizium viride.

## Data Availability

The NCBI Acc.-Nr. for the newly generated fragments of the RPB1 are PV231534, PV231535, PV231536, PV231537, PV231538, PV231539, PV231540, PV231541, PV231542, PV231543, PV231544, PV231545, PV231546, PV231547, PV231548, PV231549, PV231550, PV231551, PV231552, PV231553, PV231554, PV231555, PV231556. For RPB2, the NCBI Acc.-Nr. are PV231557, PV231558, PV231559, PV231560, PV231561, PV231562, PV231563, PV231564, PV231565, PV231566, PV231567, PV231568, PV231569, PV231570, PV231571, PV231572, PV231573, PV231574, PV231575, PV231576, PV231577, PV231578, PV231579. NCBI Acc.-Nr. for the EF1α are PV231580, PV231581, PV231582, PV231583, PV231584, PV231585, PV231586, PV231587, PV231588, PV231589, PV231590, PV231591, PV231592, PV231593, PV231594, PV231595, PV231596, PV231597, PV231598, PV231599, PV231600, PV231601, PV231602.

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
