# Peer review of "Phylogenetic Reclassification of Metarhizium granulomatis and Metarhizium viride Species Complex"

_pathogens, 2025, doi:10.3390/pathogens14080745_

Round 1
Reviewer 1 Report
Comments and Suggestions for Authors
The authors present a very interesting manuscript that proposes a reclassification of the Metarhizium granulomatis and M. viridae species complex. It is highlighted that the authors carried out a comprehensive multilocus analysis to differentiate between species complexes of the fungi evaluated. In addition, the techniques and primers that have been used worldwide to reclassify other entomopathogenic fungal complexes were used.
The article in general is well written, with the sections well established and the methodology well-structured according to the importance of the study.
Only some minor recommendations for improving the writing are presented below.
Line 66: What do you mean by preventive measures?
Line 85: Genotype E was not previously described. Legend to Figure 1 (Line 256-266): The methodology information is repetitive, since it was put as is in the materials and methods section. I suggest eliminating it and leaving only the description of what is presented in the phylogenetic tree. Same case as figure 2: 291-296.
The authors mention that there was no correlation between the genotypes and the affected animals, the isolation site or the pathological findings. However, they did not perform statistical analyzes to determine this correlation. Could you add the methodology you used to determine the absence of correlation?
Linea 327-330: When pathological findings or lesions are reported for the first time, as the authors mention in lines 327-330, it is appropriate to add photos of these findings. This would confirm the injury visually.
Author Response
Please see the attachement.

Reviewer 2 Report
Comments and Suggestions for Authors
Although the limitations of ITS/LSU sequencing were mentioned, it could be made clearer how the use of NRPB2 and EF1α represents a significant advancement compared to previous methods. There is a lack of a clear research hypothesis; although the objective is clearly stated, an explicit hypothesis formulation is missing. There are repetitions, with some information appearing both in the abstract and then again in the introduction in almost the same form (e.g., the role of ITS and LSU and the number of genotypes). Clearer transitions between sections should be added, for example between the literature review and the study objective.
Materials and Methods
Were negative samples (e.g., DNA-free controls) used in PCR? There is no mention of technical replicates for verifying the reproducibility of results. In cases of co-occurring genotypes (e.g., genotypes B and C in M. granulomatis), it should be explained how they were separated before analysis. Why was M. granulomatis genotype E omitted? A brief justification is needed. The phrase “All of the samples analyzed according described protocols” (line 100) is unclear—providing a specific protocol reference would be better. There is no information about DNA quality (e.g., Nanodrop spectrophotometry results). Was sequencing bidirectional (forward + reverse)? This is important for data reliability. It is not stated how sequence quality was verified (e.g., with software like Phred). Some procedures may require specifying the number of repetitions or replicates, e.g., in PCR or sequencing.
Results
Overall, the results are well described and demonstrate a solid approach to phylogenetic analysis. They indicate a high genetic identity of most isolated strains with reference species, confirming their classification.
Discussion
Subheadings are lacking, which could help organize the discussion. The text contains a lot of content and is sometimes repetitive, making it difficult to quickly grasp the main conclusions. For example, information about reclassification of selected genotypes appears in multiple places, which distracts the reader. Descriptions of single isolated strains with minor relevance in the context of global interpretation may distract and reduce readability. Repetitions (e.g., repeatedly discussing identical similarity values) hinder focus on key findings. Although the authors mention various clinical symptoms, they do not attempt to link specific genotypes to observed pathologies. There is some overinterpretation of genetic results; in several places, the authors suggest creating new subspecies based solely on minor sequence differences, not always supported by clear phenotypic or pathological features. The 98.7% threshold is mainly applicable to bacteria, so applying it to fungi without full justification may be an overreach. The authors acknowledge that no statistically significant correlations were found between lizard species, isolation origin, or symptoms. Unfortunately, no numerical data, statistical tests, or p-values are shown, which weakens methodological transparency. There are numerous linguistic ambiguities, excessive use of passive voice, and complex sentence structures. For example, “Either way, four genotypes of M. viride could be distinguished...” is unnecessarily colloquial and imprecise.
There is no summary.
Comments on the Quality of English LanguageThe English language needs improvement.
Round 2
Reviewer 2 Report
Comments and Suggestions for Authors
Accept in present form